# Enhancing Reasoning Capabilities of LLMs via Principled Synthetic Logic Corpus

**Terufumi Morishita**[1]      **Gaku Morio**[1*]      **Atsuki Yamaguchi**[2*†]      **Yasuhiro Sogawa**[1]

[1]Advanced AI Innovation Center, Hitachi      [2]The University of Sheffield

## Abstract

Large language models (LLMs) are capable of solving a wide range of tasks, yet they have struggled with reasoning. To address this, we propose **Additional *Logic* Training (ALT)**, which aims to enhance LLMs' reasoning capabilities by program-generated logical reasoning samples. We first establish principles for designing high-quality samples by integrating symbolic logic theory and previous empirical insights. Then, based on these principles, we construct a synthetic corpus named **Formal Logic *D*eduction *Diverse*** (FLD$_{\times 2}$), comprising numerous samples of multi-step deduction with unknown facts, diverse reasoning rules, diverse linguistic expressions, and challenging distractors. Finally, we empirically show that ALT on FLD$_{\times 2}$ substantially enhances the reasoning capabilities of state-of-the-art LLMs, including LLaMA-3.1-70B. Improvements include gains of up to 30 points on logical reasoning benchmarks, up to 10 points on math and coding benchmarks, and 5 points on the benchmark suite BBH.

## 1   Introduction

Knowledge and reasoning have long been considered essential elements for achieving *artificial intelligence* (McCarthy, 1959; Weizenbaum, 1966; Winograd, 1971; Colmerauer and Roussel, 1973; Shortliffe, 1976; Elkan and Greiner, 1993). Knowledge refers to facts about the world, e.g., "objects with mass generate a gravitational field" and "the Earth has mass." Reasoning involves combining multiple facts according to specific rules to obtain new knowledge. For example, the new knowledge that "the Earth generates a gravitational field" can be derived from the aforementioned two facts.

Recent observations suggest that LLMs can solve problems using memorized knowledge of similar samples seen during pre-training, but they cannot solve novel, unknown problems that require reasoning (Hodel and West, 2023; Dasgupta et al., 2023; Zhang et al., 2024). For instance, LLMs can solve famous arithmetic problems as is but not when the numbers or names are changed (Razeghi et al., 2022; Mirzadeh et al., 2024), and they can solve coding tests from past years before the "knowledge cutoff" but not from the present year (Mitchell, 2023). This bias towards knowledge has been observed even in state-of-the-art LLMs such as GPT-4 (Liu et al., 2023b; Wu et al., 2023; Dziri et al., 2023).

LLMs' poor reasoning capabilities can stem from the lack of high-quality reasoning samples in the pre-training corpus, which primarily consists of human-written texts (Betz et al., 2021; Morishita et al., 2023). Indeed, reasoning samples in human-written texts often exhibit low quality, as evidenced by fallacies and biases commonly found in online debates (Hansson, 2004; Guiaşu and Tindale, 2018; Cheng et al., 2017). This is unsurprising given that humans usually think reflexively rather than through rigid reasoning (Kahneman, 2011; Sunstein and Hastie, 2015; Paglieri, 2017). Thus, a

---

*Equal Contribution

†Work done at Hitachi

38th Conference on Neural Information Processing Systems (NeurIPS 2024).

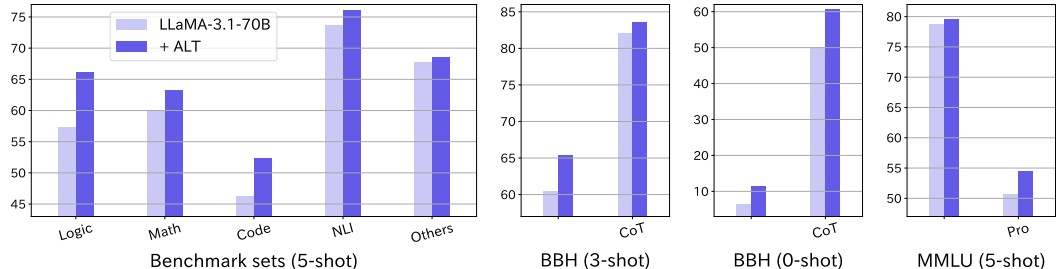

Figure 1: The performance gains to LLaMA-3.1-70B by Additional Logic Training (ALT) on the proposed synthetic corpus, FLD$_{\times 2}$ (Formal Logic *Deduction Diverse*). Each benchmark set, such as "Logic" and "Math", comprises various benchmarks in that domain. Tables 2, 4 shows the details.

straightforward strategy to improve LLMs' reasoning capabilities is to prepare many high-quality reasoning samples and train LLMs on them.

We propose one such approach, **Additional *Logic* Training (ALT)**, which utilizes high-quality samples of *logical* reasoning, the most fundamental form of reasoning. To prepare such samples, we utilize synthetic generation (Clark et al., 2021; Betz et al., 2021; Tafjord et al., 2021; Morishita et al., 2023), where computer programs generate deductive reasoning samples in which a given hypothesis is proven or disproven by combining given facts following rigid reasoning rules. We overview ALT in Figure 2.

In synthetic generation, computer programs generate samples according to pre-designed patterns, so this design largely determines the quality of these samples by nature. Thus, we start by discussing **what is the ideal design for synthetic logic samples**, incorporating symbolic logic theory and empirical findings (Section 2). The essence of logical reasoning lies in its ability to handle unknown facts, unlike knowledge, which deals solely with established facts, such as commonsense facts; therefore, samples must cover reasoning with unknown facts. Samples must include both *illogical* and logical reasoning to enable LLMs to distinguish between them. The samples must cover various patterns regarding a comprehensive set of reasoning aspects, such as reasoning rules and linguistic expressions of logical statements. We summarize these discussions into *design principles*, which guide the design of synthetic logic samples. Finally, based on these principles, we construct a synthetic corpus named **Formal Logic *Deduction Diverse*** (FLD$_{\times 2}$), comprising numerous samples of multi-step deduction with unknown facts, diverse reasoning rules, diverse linguistic expressions, and challenging distractors (Section 3).

We then empirically verify that ALT can enhance LLMs' reasoning capabilities (Sections 4, 5). Using 31 benchmarks covering diverse tasks, we observed that ALT on FLD$_{\times 2}$ substantially boosts state-of-the-art LLMs' reasoning capabilities. Even LLaMA-3.1-70B, the largest LLM pre-trained on over 15 trillion tokens, shows substantial improvements with ALT (Figure 1). Among synthetic logic corpora with different sample designs, FLD$_{\times 2}$ yielded the largest performance gains, validating our proposed design principles. Moreover, we discovered that employing a knowledge-forgetting prevention method during ALT is critically important, as it likely prevents the LLM's knowledge of established facts from being displaced by the unknown facts included in synthetic logic corpora.

Finally, we analyze which task-solving capabilities ALT can enhance and why (Section 6). We observed a substantial improvement of up to 30 points on logical reasoning tasks (Table 4a). Surprisingly, we also observed improvements in abductive reasoning tasks, which go beyond the synthetic logic corpora's original deductive reasoning tasks. Case analyses indicate that these improvements result from LLMs having acquired the fundamentals of the logic reflected in the design principles. We also observed improvements of up to 10 points on math and coding tasks, indicating the generalizability of the obtained reasoning capabilities (Tables 4b, 4c). We also observed improvements of up to 6 points on natural language inference (NLI) tasks (Table 4d). Case analyses suggest that LLMs successfully integrated the commonsense knowledge they had originally acquired during pre-training with the logical reasoning capabilities newly acquired from ALT.

Improvements across various other tasks (Table 4e) demonstrate the broad benefits of the obtained reasoning capabilities beyond standard reasoning tasks, though the modest improvements of up to 2 points indicate the need for future research on more effective application of these capabilities.

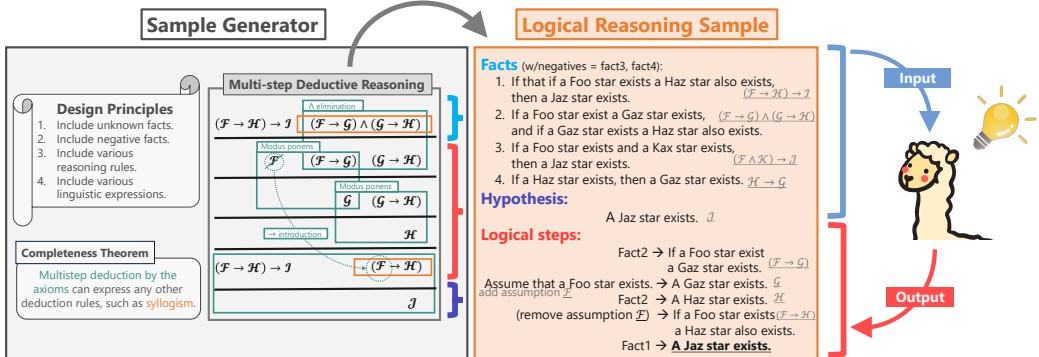

Figure 2: Our proposed **A**dditional **L**ogic **T**raining (ALT) aims to enhance LLMs' reasoning capabilities through training on many synthetically generated logical reasoning samples. Our sample generator (left) first generates a sample of multi-step deductive reasoning and then converts it into a deduction sample written in English (right). LLMs must generate **logical steps** to derive a given **hypothesis** from provided **facts**. The sample generator adheres to theoretically and empirically grounded *design principles* discussed in Section 2. Refer to Figure D.3 for a real sample.

Our contributions are summarized as follows:

- We propose **A**dditional ***L****ogic* **T**raining (ALT) and empirically verify that it can enhance the reasoning capability of state-of-the-art LLMs across various sizes, from 7B to 70B.

- We establish systematic design principles for synthetic logic samples; then, we construct a synthetic corpus named **Formal Logic *Deduction Diverse*** (FLD$_{\times 2}$), comprising numerous samples of multi-step deduction with unknown facts, diverse reasoning rules, diverse linguistic expressions, and challenging distractors. We empirically verify that Formal Logic Deduction Diverse indeed leads to the largest improvements among corpora with different sample designs.

- We demonstrate that LLMs enhanced by ALT can solve not only the original logical reasoning tasks present in synthetic logic corpora but also other tasks, such as math and coding tasks, and notably NLI tasks, which require integrating knowledge and reasoning. This finding underscores the potential for advancing truly versatile AI possessing both knowledge and reasoning capabilities.

We release the corpus, code, and the trained model under a permissive license [1].

## 2 How Should Synthetic Logic Samples Be Designed?

In synthetic generation, computer programs generate samples according to pre-designed patterns, so this design largely determines the quality of the samples. While Previous studies have examined several designs (Clark et al., 2021; Betz et al., 2021; Tafjord et al., 2021; Morishita et al., 2023), these designs were not systematically discussed, so they may not be the most effective ones.

Thus, we start by discussing how to optimally design synthetic logic samples. To this end, we consider symbolic logic theory as suggested by Morishita et al. (2023) and integrate empirical findings from previous studies. First, we observe that the essence of logical reasoning, based solely on the logical relationships between facts, lies in its ability to handle unknown facts, unlike knowledge, which by definition deals solely with established facts (Section 2.1). Therefore, we argue that samples should cover reasoning with unknown facts to represent this essential aspect of logical reasoning. We also observe that logical reasoning involves various other aspects, such as *illogical* reasoning, reasoning rules, and linguistic expressions that represent logical statements (sections 2.2 to 2.4). The samples should cover various patterns regarding these aspects to enable LLMs to solve various reasoning problems. We summarize these discussions into the following *design principles*, which guide the design of synthetic logic samples.

---

[1] https://github.com/hitachi-nlp/FLD

## 2.1 Teaching Reasoning with Unknown Facts

We first explore the essence of logical reasoning that differentiates itself from knowledge. Consider the following logical step:

$$\frac{\text{The Earth orbits the Sun.} \qquad \text{If the Earth orbits the sun, the Earth has four seasons.}}{\text{The Earth has four seasons.}} \tag{1}$$

This step is valid because the conclusion is logically derived from the two premises. Next, consider another logical step:

$$\frac{\text{The Earth orbits the Sun.} \qquad \text{If the Earth orbits the sun, the Earth } \textit{does not have } \text{four seasons.}}{\text{The Earth } \textit{does not have } \text{four seasons.}} \tag{2}$$

The second premise and consequently, the conclusion, is factually wrong. Nevertheless, *if the premise was hypothetically correct*, the conclusion could be logically derived. Therefore, step (2) is also logically valid. Finally:

$$\frac{\text{1. A Foo star exists.} \qquad \text{2. If a Foo star exists, a Bar star also exists.}}{\text{A Bar star exists.}} \tag{3}$$

"Foo star" and "Bar star" are unknowns; nonetheless, we can still determine that step (3) is logically valid. Steps (1) to (3) above can be abstracted into a **deduction rule**, i.e., modus ponens, using symbols:

$$\frac{\mathcal{F} \qquad \mathcal{F} \to \mathcal{G}}{\mathcal{G}} \text{ modus ponens} \tag{4}$$

As we have seen, the logical validity of a deduction rule depends solely on whether the conclusion is logically derived from the premises, not on the factual correctness of the contents of $\mathcal{F}$ and $\mathcal{G}$. Therefore, *the contents of $\mathcal{F}$ and $\mathcal{G}$ can be arbitrary.*

Now, we consider what kind of samples would be needed to teach the deduction rule (4) to LLMs. We assume a task to generate the conclusion given the premises as prompt inputs. If the learner were human, they would be able to infer the underlying deduction rule (4) by observing samples such as (1) to (2). As a result, they would become able to solve the unknown problem (3).

However, from a purely inductive perspective, samples (1) to (2) cannot simply be generalized to the deduction rule (4). This is because the samples (1) to (2) themselves do not contain the information that the contents of $\mathcal{F}$ and $\mathcal{G}$ are arbitrary. In fact, one could generalize samples (1) to (2) to other rules; for example, the conclusion $\mathcal{G}$ can be derived if $\mathcal{F}$ and $\mathcal{F} \to \mathcal{G}$ are given as premises *and $\mathcal{F}$ and $\mathcal{G}$ include 'Earth' as their contents*. Innumerable such deduction rules can be inductively inferred from the given samples. In other words, induction has arbitrariness (Hume, 1748; Goodman, 1954; Quine, 1969).

Humans prefer simpler rules (Bertrand; Wittgenstein, 1922), so they boldly induce up to the deduction rule (4). However, it is unclear how purely inductive learners such as LLMs, which extract only what can be inferred from samples without prior preferences, induce up to (4). For example, if only specific contents such as "Alice is kind" and "Bob is smart" are assigned to $\mathcal{F}$ and $\mathcal{G}$ in training samples, an LLM could develop into a machine that generates the conclusion $\mathcal{G}$ only when the input contains the specific contents. In order for LLMs to accurately induce that $\mathcal{F}$ and $\mathcal{G}$ are indeed arbitrary:

**Design Principle 1** (Reasoning with Unknown Facts). *Prepare many samples assigning arbitrary contents to $\mathcal{F}$ and $\mathcal{G}$. They will make LLMs accurately induce $\mathcal{F}$ and $\mathcal{G}$ are indeed arbitrary, ultimately enabling them to reason with unknown facts.*

## 2.2 Teaching Illogical Reasoning

Suppose we have LLMs trained on a large number of samples as follows:

$$\frac{\mathcal{F} \wedge \mathcal{G} \qquad (\mathcal{F} \wedge \mathcal{G}) \to \mathcal{H}}{\mathcal{H}} \tag{5}$$

where $\wedge$ denotes logical conjunction, and arbitrary contents are assigned to $\mathcal{F}, \mathcal{G}, \mathcal{H}$. Suppose that we give this LLM a problem such as:

$$\frac{\mathcal{F} \qquad (\mathcal{F} \wedge \mathcal{G}) \to \mathcal{H}}{??} \tag{6}$$

Since the premises are insufficient for logically deducting the conclusion, outputting nothing is the correct answer.

Unfortunately, an LLM could output $\mathcal{H}$, which was indeed often observed in our preliminary experiments. This is because while the LLMs can induce from sample (5) that it can generate the conclusion $\mathcal{H}$ when the two premises of (5) are given, the LLMs cannot induce from the sample that it is *not allowed* to generate the conclusion $\mathcal{H}$ when the premises of (6) are given, as such information is not included in the sample (5) itself. Therefore,

**Design Principle 2** (Illogical Reasoning). *Include negative samples such as (6). These samples will make LLMs induce that conclusions cannot be derived from insufficient premises.*

## 2.3 Teaching Diverse Reasoning Rules

Deduction rules other than (4) exist:

$$\frac{(\mathcal{F}\wedge\mathcal{G})}{\mathcal{F}} \quad \frac{(\mathcal{F}\wedge\mathcal{G})}{\mathcal{G}} \ \wedge\text{elimination} \qquad \frac{(\mathcal{F}\rightarrow\mathcal{G})\wedge(\mathcal{G}\rightarrow\mathcal{H})}{\mathcal{F}\rightarrow\mathcal{H}} \ \text{syllogism}$$

$$\frac{\mathcal{F}\rightarrow\mathcal{G}}{\neg\mathcal{G}\rightarrow\neg\mathcal{F}} \ \text{contraposition} \qquad \frac{\neg(\mathcal{F}\vee\mathcal{G})}{\neg\mathcal{F}\wedge\neg\mathcal{G}} \quad \frac{\neg(\mathcal{F}\wedge\mathcal{G})}{\neg\mathcal{F}\vee\neg\mathcal{G}} \ \text{De Morgan's laws} \tag{7}$$

where $\vee$ denotes logical disjunction and $\neg$ negation. Since there are infinitely many possible logical formulas that can appear as premises and conclusions, there are infinitely many deduction rules. Providing LLMs with these infinite deduction rules is obviously intractable.

Instead of directly providing these infinite deduction rules, we can take another approach. Consider multi-step deductive reasoning (Figure 2 left), where multiple deduction rules derive a conclusion. Notice that the syllogism in (7) can be expressed by multi-step deductive reasoning using more "atomic" deduction rules. Indeed, there exists a set of atomic deduction rules called **the axioms** that satisfies the following:

**Theorem 2.1** (Completeness of First-Order Predicate Logic Gödel (1930)). *Any valid deduction rule can be expressed by multistep deductive reasoning constructed from the axioms.*

In contrast to the axioms, the 'compound' deduction rules, such as syllogism, contraposition, and De Morgan's laws, are called theorems. According to the completeness Theorem 2.1, if we can handle the axioms, we can effectively handle other deduction rules as well. Indeed, Morishita et al. (2023) empirically verified that a language model trained on the axioms generalizes to handle other deduction rules more effectively than those trained on non-axiom deduction rules. Therefore,

**Design Principle 3** (Diverse Reasoning Rules). *Samples should express multi-step deduction constructed from the axioms. They will effectively teach LLMs diverse deduction rules (Morishita et al., 2023)*

In multi-step deductive reasoning, the number of logical steps $s$ from premises to a conclusion can vary largely depending on the problem. Therefore:

**Design Principle 3'** (Diverse Reasoning Rules). *Samples should include diverse numbers of logical steps $s$.*

Ideally, this would be sufficient, but empirical evidence has shown that LLMs struggle with constructing multi-step deductive reasoning with large steps $s$ (Gontier et al., 2020; Morishita et al., 2023). Consequently, LLMs would not excel at handling theorems that require a large number of steps $s$ when expressed by the axioms. Therefore, as an additional countermeasure:

**Design Principle 3"** (Diverse Reasoning Rules). *Samples should also include representative theorems, such as syllogism, contraposition, and De Morgan's laws.*

## 2.4 Teaching Diverse Linguistic Expressions that Represent Logical Statements

There are various linguistic structures for expressing the logical relationship $\mathcal{F}\rightarrow\mathcal{G}$, such as "If $\mathcal{F}$ then $\mathcal{G}$", "$\mathcal{F}$ leads to $\mathcal{G}$", and "$\mathcal{F}$ results in $\mathcal{G}$". If we only include specific expressions in the corpora, LLMs may only learn to react to these specific expressions, which has been observed in previous experiments (Zhang et al., 2022; Yuan et al., 2023). To prevent this,

**Design Principle 4** (Diverse Linguistic Expressions). *Samples should include diverse linguistic expressions that represent logical statements.*

In this chapter, we have established the principles to guide the design of synthetic logic samples. Next, we construct a synthetic logic corpus based on these principles.

Table 1: Synthetic logic corpora compared in this study, with their features categorized according to our proposed design principles (DP). Note that the last row of the *ablation* corpora lists variations of $\text{FLD}_{\times 2}$, each of which differs from the original regarding one of the design principles.

| | DP1 | DP2 | DP3 | | DP4 |
|---|---|---|---|---|---|
| | vocabulary size | distractors | deduction rules | logical steps | expressions per formula |
| RuleTaker (Clark et al., 2021) (RT) | $\leq 100$ (hand-selected) | random formula | 2 (implication) | 1–5 | $\mathcal{O}(1)$ |
| PARARULE-Plus (Bao et al., 2022) (PRP) | $\leq 100$ (hand-selected) | random formula | 2 (implication) | 1–5 | $\mathcal{O}(1)$ |
| FLD (Morishita et al., 2023) | $\simeq 15\text{k}$ (WordNet, subset) | random formula | 13 (axioms) | **1–8** | 10∼100 |
| **$\text{FLD}_{\times 2}$** | $\simeq \mathbf{100k}$ (WordNet, full) | **adversarial formula** | $\simeq \mathbf{50}$ (axioms and theorems) | **1–8** | **10∼100** (more extensive than FLD) |
| $\text{FLD}_{\times 2}$ *ablation* corpora → | 100 → *w/o DP1* | not used → *w/o DP2* | 2 (implication) → *w/o DP3.rules* | 1 → *w/o DP3.steps* | 1 → *w/o DP4* |

## 3 Creating a Synthetic Corpus based on Design Principles

To prepare diverse samples reflecting the design principles 1 to 4 (DP1-4), we built a novel sample generator by extending the previous one by Morishita et al. (2023) and then generated the synthetic logic corpus named $\text{FLD}_{\times 2}$ (Formal Logic *D*eduction *D*iverse). Figure 2 shows a schematic of our generator and a deduction sample. Table 1 compares $\text{FLD}_{\times 2}$ with existing corpora. Figure D.3 provides an actual deduction sample included in $\text{FLD}_{\times 2}$.

More specifically, our generator generates deduction samples through the following steps. First, the generator randomly generates a sample of multi-step deductive reasoning written in logical formulas, as shown on the left side of Figure 2, where a conclusion is derived from premises using multiple **deduction rules** (See Appendix D.3 for more details of this generation procedure). At this time, the generator also generates 'distractor' logical formulas, which express negative premises of DP2. Next, the generator converts each logical formula into English expressions. To achieve this, the generator first randomly selects a template from pre-defined options, such as "If $\mathcal{F}$, then $\mathcal{G}$," "$\mathcal{F}$ leads to $\mathcal{G}$," or "$\mathcal{F}$ results in $\mathcal{G}$," for the logical formula "$\mathcal{F} \rightarrow \mathcal{G}$." It then assigns English content randomly constructed from a vocabulary, such as "(that) a Foo star exists" and "(that) a Bar star exists," to each symbol, such as $\mathcal{F}$ and $\mathcal{G}$. Finally, it converts the multi-step deduction into a deduction sample (right side of Figure 2) by using the premises as **'facts'**, the conclusion as **'hypothesis'**, and the intermediate logical steps as **'logical steps'**. The deduction sample requires LLMs to generate **logical steps** that derive a given **hypothesis** based on the given **facts**.

Table 1 outlines the comparison of $\text{FLD}_{\times 2}$ with other existing corpora (Clark et al., 2021; Bao et al., 2022; Morishita et al., 2023) in terms of DP1-4, which is detailed as follows:

- DP1: We assign $\mathcal{F}$ and $\mathcal{G}$ content randomly constructed from a vocabulary. While the existing corpora used small-sized vocabulary of up to 15k, we use a large vocabulary of around 100k words built from WordNet (Miller, 1995). This will teach LLMs that $\mathcal{F}$ and $\mathcal{G}$ are truly arbitrary, ultimately enabling them to reason with unknown facts.

- DP2: The existing corpora used randomly generated logical formulas as distractors. In contrast, we implement adversarial distractors. For example, for a premise $\mathcal{F} \wedge \mathcal{G}$, we use $\mathcal{F}$ with missing information (see Equations (5), (6)), and for a premise $\mathcal{F} \rightarrow \mathcal{H}$, we use $\mathcal{F} \wedge \mathcal{G} \rightarrow \mathcal{H}$ with missing information as distractors. These distractors teach LLMs precisely when a conclusion can and cannot be derived. As with previous corpora, we include a variable number of distractors in each sample, randomly chosen from a range of 0 to 20.

- DP3-3": While the existing corpora used a small number of deduction rules of up to 13 (refer to Figure B.4 of Morishita et al. (2023)), we include diverse deduction rules, encompassing the axioms and representative theorems, such as modus ponens, syllogisms, and contraposition, totaling about 50 rules. We include samples with up to $s = 8$ logical steps, following (Morishita et al., 2023).

- DP4: We manually craft several more English templates *per* logical formulas than those used in FLD. Since the templates have a nested structure, they yield combinatorially more diverse English expressions. While counting the exact number of the resulting expressions is intractable, we observed at least dozens of expressions per logical formula, including minor variations. See Appendix D.4 for details.

# 4 Experimental Setup

We briefly explain the experimental settings. Refer to Appendix E for the details.

**Synthetic Logic Corpora:** We examine the proposed $FLD_{\times 2}$ and previous corpora (Table 1).

**LLMs:** We used the state-of-the-art LLM, LLaMA-3.1 (8B and 70B) (AI@Meta, 2024).

**Training Settings:** We trained the LLMs by a method similar to supervised fine-tuning; as illustrated in Figure 2, we used the facts and hypothesis as inputs and logical steps and additional answer label (see Appendix D.1) as outputs. We excluded loss computation for the inputs to prevent LLMs from learning to generate unknown facts. We trained the LLMs for 1 epoch on 100k samples ($\sim 0.1B$ tokens) from the training split of each corpus, with a batch size of 256, resulting in 390 steps, with a linear warmup for 200 steps. We used the learning rate of 2e-05 for the 8B model and 3e-06 for the 70B model. We used Huggingface (Wolf et al., 2020) for implementation.

**Prevention of Knowledge Forgetting by Recall Adam Optimizer:** Synthetic logic corpora include many samples with unknown facts, so training on them should cause LLMs to forget their knowledge of existing facts. To prevent this, we employed the Recall Adam optimizer (Chen et al., 2020), which regularizes parameter updates to avoid deviating too far from the pre-training parameters. Recall Adam stands out for LLM training for several reasons (see Appendix E.0.1 for details). We used our re-implemented version [2]. The hyperparameters were: $\beta_1 = 0.9, \beta_2 = 0.999, \epsilon = 10^{-6}$, fisher coefficient $= 4000$ for the 8B model and 2000 for the 70B model.

**Benchmarks:** We evaluated the trained LLMs on 31 benchmarks shown in Table E.7 using 5-shot in-context learning, except for BBH and AbductionRules, which used 3-shot in-context learning. These benchmarks cover a wide range of tasks and are prominent in LLM evaluation. Note that we excluded the synthetic logic corpora used for training, as training on them often leads to overfitting to their superficial and statistical cues (Zhang et al., 2022; Yuan et al., 2023), failing to measure truly generalizable reasoning capabilities. We used lm-evaluation-harness (Gao et al., 2023) and bigcode-evaluation-harness (Ben Allal et al., 2022) for the implementation.

# 5 Can Additional Logic Training Enhance LLMs' Capabilities?

Table 2 show the performance of LLMs before and after ALT. Most LLMs trained with ALT outperformed their counterparts without ALT. Notably, ALT yielded substantial gains of up to 10 points even for LLaMA-3.1-70B, the largest LLM pre-trained on over 15 trillion tokens. These results verify that ALT can enhance the capabilities of state-of-the-art LLMs.

Among the LLMs trained with ALT, the one trained on $FLD_{\times 2}$ (i.e., $\oplus \textbf{ALT}\text{-}FLD_{\times 2}$) achieved the highest generalization performance across the benchmarks. Table 3 shows the performance of the LLMs trained on *ablated* $FLD_{\times 2}$ corpora, each of which lacks one of the design principles. As seen, ablating any design principle almost always led to performance degradation. These results demonstrate that the proposed design principles are critical to obtaining the maximum possible gain from ALT, and each principle is indispensable.

Table F.8 shows that the LLMs trained with ALT without preventing knowledge forgetting by Recall Adam optimizer underperformed compared to their counterparts trained with knowledge forgetting prevention and even the LLM without ALT. This behavior presumably occurred because the unknown facts included in synthetic logic corpora displaced the LLM's knowledge of existing facts. Therefore, knowledge-forgetting prevention is critically important for the success of ALT.

# 6 What Capabilities Can Additional Logic Training Enhance and Why?

We analyze the results on each benchmark or each case and discuss whether and why the LLM's capabilities to solve the tasks can or cannot be enhanced by ALT.

## 6.1 Logical Reasoning Tasks

Table 4a shows that ALT substantially boosted LLaMA-3.1-70B's performance by up to 30 points on various benchmarks dealing with logical reasoning tasks. Surprisingly, we also observed improvements on abductive reasoning tasks, which go beyond the original deductive reasoning tasks

---

[2] https://github.com/hitachi-nlp/rec-adam

Table 2: 5-shot performance of LLMs before and after ALT. ⊕**ALT**-$x$ denotes the LLM trained with ALT on the synthetic logic corpus $x$ from Table 1. The color shows the rank in each column (darker is better). Each benchmark set, such as "Logic" and "Math", comprises various benchmarks in that domain (see Table E.7). "Avg." represents the micro-average of all the benchmarks.

(a) LLaMA-3.1-8B.

| | Avg. | Logic | Math | Code | NLI | Others | BBH (3-shot) | | BBH (0-shot) | | MMLU | |
| --- | --- | --- | --- | --- | --- | --- | --- | --- | --- | --- | --- | --- |
| | | | | | | | | CoT | | CoT | | Pro |
| LLaMA-3.1-8B | 47.9 | $42.8_{\pm0.4}$ | $39.6_{\pm0.5}$ | 35.4 | $65.4_{\pm0.3}$ | $60.7_{\pm0.3}$ | $44.9_{\pm0.4}$ | $61.9_{\pm0.4}$ | $8.2_{\pm0.2}$ | $36.5_{\pm0.4}$ | $65.3_{\pm0.4}$ | $35.8_{\pm0.4}$ |
| ⊕ALT-PRP | 48.1 | $43.7_{\pm0.2}$ | $39.2_{\pm0.3}$ | 35.7 | $65.6_{\pm0.2}$ | $60.8_{\pm0.2}$ | $44.9_{\pm0.2}$ | $61.8_{\pm0.2}$ | $8.2_{\pm0.1}$ | $36.4_{\pm0.2}$ | $65.3_{\pm0.2}$ | $35.3_{\pm0.2}$ |
| ⊕ALT-RT | 50.1 | $46.8_{\pm0.1}$ | $42.4_{\pm0.2}$ | 36.5 | $68.6_{\pm0.1}$ | $61.3_{\pm0.2}$ | $46.9_{\pm0.2}$ | $63.5_{\pm0.2}$ | $13.7_{\pm0.1}$ | $38.4_{\pm0.2}$ | $65.3_{\pm0.1}$ | $35.7_{\pm0.2}$ |
| ⊕ALT-FLD | 51.9 | $51.6_{\pm0.1}$ | $43.4_{\pm0.2}$ | 38.1 | $70.1_{\pm0.1}$ | $61.5_{\pm0.1}$ | $46.7_{\pm0.2}$ | $64.9_{\pm0.2}$ | $11.9_{\pm0.1}$ | $39.6_{\pm0.2}$ | $65.4_{\pm0.1}$ | $36.2_{\pm0.2}$ |
| ⊕**ALT-FLD**$_{\times2}$ | 52.0 | $52.2_{\pm0.1}$ | $43.2_{\pm0.2}$ | 38.0 | $70.7_{\pm0.1}$ | $61.5_{\pm0.1}$ | $46.5_{\pm0.2}$ | $65.3_{\pm0.2}$ | $11.3_{\pm0.1}$ | $38.7_{\pm0.2}$ | $65.5_{\pm0.1}$ | $36.4_{\pm0.2}$ |

(b) LLaMA-3.1-70B.

| | Avg. | Logic | Math | Code | NLI | Others | BBH (3-shot) | | BBH (0-shot) | | MMLU | |
| --- | --- | --- | --- | --- | --- | --- | --- | --- | --- | --- | --- | --- |
| | | | | | | | | CoT | | CoT | | Pro |
| LLaMA-3.1-70B | 60.0 | $57.4_{\pm0.4}$ | $60.0_{\pm0.5}$ | 46.2 | $73.7_{\pm0.3}$ | $67.7_{\pm0.3}$ | $60.4_{\pm0.3}$ | $82.1_{\pm0.2}$ | $6.5_{\pm0.1}$ | $50.1_{\pm0.3}$ | $78.7_{\pm0.3}$ | $50.7_{\pm0.4}$ |
| ⊕ALT-PRP | 60.4 | $57.7_{\pm0.4}$ | $59.8_{\pm0.5}$ | 49.2 | $73.5_{\pm0.3}$ | $67.6_{\pm0.3}$ | $60.4_{\pm0.4}$ | $82.2_{\pm0.3}$ | $6.0_{\pm0.2}$ | $50.1_{\pm0.4}$ | $78.7_{\pm0.3}$ | $50.9_{\pm0.4}$ |
| ⊕ALT-RT | 62.7 | $61.4_{\pm0.2}$ | $62.1_{\pm0.3}$ | 50.8 | $75.4_{\pm0.2}$ | $68.4_{\pm0.2}$ | $64.1_{\pm0.3}$ | $82.5_{\pm0.2}$ | $11.5_{\pm0.2}$ | $59.2_{\pm0.3}$ | $79.0_{\pm0.2}$ | $52.4_{\pm0.3}$ |
| ⊕ALT-FLD | 64.2 | $65.7_{\pm0.1}$ | $63.6_{\pm0.2}$ | 52.0 | $75.3_{\pm0.1}$ | $68.5_{\pm0.1}$ | $65.0_{\pm0.2}$ | $83.6_{\pm0.1}$ | $12.1_{\pm0.1}$ | $59.9_{\pm0.2}$ | $79.3_{\pm0.1}$ | $54.4_{\pm0.2}$ |
| ⊕**ALT-FLD**$_{\times2}$ | 64.4 | $66.1_{\pm0.1}$ | $63.3_{\pm0.2}$ | 52.4 | $76.1_{\pm0.1}$ | $68.5_{\pm0.1}$ | $65.4_{\pm0.2}$ | $83.6_{\pm0.2}$ | $11.4_{\pm0.1}$ | $60.8_{\pm0.2}$ | $79.5_{\pm0.1}$ | $54.4_{\pm0.2}$ |

Table 3: LLaMA-3.1-8B trained on the ablation corpora.

| | Avg. | Logic | Math | Code | NLI | Others | BBH (3-shot) | | BBH (0-shot) | | MMLU | |
| --- | --- | --- | --- | --- | --- | --- | --- | --- | --- | --- | --- | --- |
| | | | | | | | | CoT | | CoT | | Pro |
| ⊕**ALT-FLD**$_{\times2}$ | 52.0 | $52.2_{\pm0.1}$ | $43.2_{\pm0.2}$ | 38.0 | $70.7_{\pm0.1}$ | $61.5_{\pm0.1}$ | $46.5_{\pm0.2}$ | $65.3_{\pm0.2}$ | $11.3_{\pm0.1}$ | $38.7_{\pm0.2}$ | $65.5_{\pm0.1}$ | $36.4_{\pm0.2}$ |
| w/o DP1 | 51.4 | $52.2_{\pm0.1}$ | $43.1_{\pm0.1}$ | 39.2 | $70.0_{\pm0.1}$ | $59.4_{\pm0.1}$ | $46.7_{\pm0.2}$ | $64.7_{\pm0.2}$ | $11.5_{\pm0.1}$ | $38.9_{\pm0.2}$ | $65.4_{\pm0.1}$ | $36.1_{\pm0.2}$ |
| w/o DP2 | 50.6 | $49.9_{\pm0.1}$ | $43.1_{\pm0.1}$ | 38.1 | $71.1_{\pm0.1}$ | $59.3_{\pm0.1}$ | $46.1_{\pm0.2}$ | $64.6_{\pm0.2}$ | $10.4_{\pm0.1}$ | $37.4_{\pm0.2}$ | $65.4_{\pm0.1}$ | $35.7_{\pm0.2}$ |
| w/o DP3.rules | 50.7 | $50.4_{\pm0.1}$ | $42.8_{\pm0.2}$ | 38.3 | $69.5_{\pm0.1}$ | $59.4_{\pm0.1}$ | $46.4_{\pm0.2}$ | $64.0_{\pm0.2}$ | $11.8_{\pm0.1}$ | $38.3_{\pm0.2}$ | $65.6_{\pm0.1}$ | $36.2_{\pm0.2}$ |
| w/o DP3.steps | 51.1 | $51.5_{\pm0.1}$ | $43.1_{\pm0.2}$ | 38.7 | $69.6_{\pm0.1}$ | $59.5_{\pm0.1}$ | $46.8_{\pm0.2}$ | $65.0_{\pm0.2}$ | $12.3_{\pm0.1}$ | $38.8_{\pm0.2}$ | $65.6_{\pm0.1}$ | $36.3_{\pm0.2}$ |
| w/o DP4 | 51.3 | $52.2_{\pm0.1}$ | $42.8_{\pm0.2}$ | 38.4 | $70.3_{\pm0.1}$ | $59.5_{\pm0.1}$ | $46.1_{\pm0.2}$ | $64.8_{\pm0.2}$ | $12.8_{\pm0.1}$ | $39.3_{\pm0.2}$ | $65.5_{\pm0.1}$ | $36.3_{\pm0.2}$ |

in synthetic logic corpora. Abductive reasoning involves guessing the missing premises that caused the observed conclusion rather than deriving a conclusion from the premises. For example, from the observed conclusion, "the window glass at home was broken and the room was ransacked," we guess the premise "a burglar broke in." The improvements would be due to the fact that, while the surface form of abductive reasoning problems differs from that of deductive reasoning, they share the fundamentals of logic reflected in the design principles.

Next, we conduct case analyses to see whether the LLM enhanced by ALT acquired the abilities intended by the proposed design principles (DP1-4). Table 5 shows problems where LLaMA-3.1-70B's errors have been corrected by ALT. The first problem is very simple, so it is surprising that LLaMA-3.1-70B failed to solve it, indicating the inherent difficulty of learning logical reasoning solely from pre-training. In contrast, ⊕ALT-FLD$_{\times2}$, which was additionally trained on FLD$_{\times2}$, solved the problem correctly. The premises of the problem are randomly constructed to express unknown facts. Therefore, the result suggests that ⊕ALT-FLD$_{\times2}$ acquired genuine logical reasoning ability, which can handle unknown facts (DP1).

In the second problem, ⊕ALT-FLD$_{\times2}$ correctly answered "neutral", indicating that it successfully learned that conclusions cannot be derived from insufficient facts (DP2).

The third problem comes from the FOLIO benchmark. To solve this problem, LLMs must use syllogism at the first step as follows: "All eels are fish, and no fish are plants. Therefore, all ells are not plants." ⊕ALT-FLD$_{\times2}$ answered this problem correctly, suggesting that it successfully learned diverse deduction rules (DP3).

FOLIO problems are created based on Wikipedia topics, describing them in more natural and realistic linguistic expressions than in other benchmarks. As seen in the fourth problem, ⊕ALT-FLD$_{\times2}$ understands such expressions, suggesting the effect of diverse expressions from DP4 and/or that LLMs can integrate their original linguistic ability with the newly acquired logical reasoning ability.

Table 4: Benchmark-wise 5-shot performance of LLaMA-3.1-70B before and **after** ALT on FLD$_{\times2}$. Refer to Table F.9 for LLaMA-3.1-8B results. Table E.7 details each benchmark.

(a) Logic.

| | bAbiD | FOLIO | LogicNLI | RobustLR | AR-LSAT | LogiQA | ReClor | AbductionR | ART |
|---|---|---|---|---|---|---|---|---|---|
| LLaMA-3.1-70B | $83.8_{\pm1.2}$ | $58.9_{\pm1.6}$ | $34.9_{\pm1.1}$ | $49.6_{\pm0.9}$ | $21.5_{\pm1.0}$ | $64.3_{\pm1.2}$ | $33.7_{\pm0.7}$ | $84.0_{\pm0.7}$ | $85.4_{\pm0.9}$ |
| ⊕**ALT-FLD**$_{\times2}$ | $83.5_{\pm0.5}$ | $\mathbf{66.7}_{\pm0.6}$ | $\mathbf{50.9}_{\pm0.5}$ | $\mathbf{81.6}_{\pm0.3}$ | $\mathbf{25.0}_{\pm0.4}$ | $\mathbf{69.4}_{\pm0.5}$ | $\mathbf{36.3}_{\pm0.3}$ | $\mathbf{95.7}_{\pm0.2}$ | $\mathbf{85.5}_{\pm0.4}$ |

(b) Math.

| | GSM8k | | | MATH | MathQA |
|---|---|---|---|---|---|
| | CoT | CoT (0-shot) | | - | - |
| LLaMA-3.1-70B | $80.9_{\pm1.1}$ | $75.2_{\pm1.2}$ | $65.4_{\pm1.3}$ | $23.7_{\pm0.6}$ | $55.0_{\pm0.9}$ |
| ⊕**ALT-FLD**$_{\times2}$ | $\mathbf{83.3}_{\pm0.4}$ | $\mathbf{80.4}_{\pm0.4}$ | $\mathbf{73.0}_{\pm0.5}$ | $\mathbf{24.4}_{\pm0.2}$ | $\mathbf{55.4}_{\pm0.4}$ |

(c) Code.

| | HumanEval | MBPP | MBPP+ | MultiPL-E (cpp) | MultiPL-E (go) |
|---|---|---|---|---|---|
| LLaMA-3.1-70B | 32.3 | 43.4 | 48.7 | 29.8 | 76.6 |
| ⊕**ALT-FLD**$_{\times2}$ | **42.6** | **49.5** | **52.5** | **38.7** | **78.6** |

(d) Natural language inference (NLI).

| | HELP | MNLI | RTE | SNLI |
|---|---|---|---|---|
| LLaMA-3.1-70B | $45.8_{\pm0.5}$ | $82.2_{\pm0.4}$ | $84.0_{\pm0.7}$ | $\mathbf{82.6}_{\pm0.4}$ |
| ⊕**ALT-FLD**$_{\times2}$ | $\mathbf{51.3}_{\pm0.2}$ | $\mathbf{83.7}_{\pm0.2}$ | $\mathbf{87.2}_{\pm0.3}$ | $82.3_{\pm0.2}$ |

(e) Others.

| | CommonsenseQA | HellaSwag | SQuAD | WinoGrande | ARCe | ARCc | GPQA | OpenBookQA | SciQ |
|---|---|---|---|---|---|---|---|---|---|
| LLaMA-3.1-70B | $81.2_{\pm1.1}$ | $69.2_{\pm0.5}$ | $38.5_{\pm0.0}$ | $85.6_{\pm1.0}$ | $89.1_{\pm0.6}$ | $65.3_{\pm1.4}$ | $\mathbf{40.7}_{\pm1.4}$ | $41.4_{\pm0.7}$ | $98.5_{\pm0.4}$ |
| ⊕**ALT-FLD**$_{\times2}$ | $\mathbf{82.5}_{\pm0.4}$ | $\mathbf{69.6}_{\pm0.2}$ | $\mathbf{40.1}_{\pm0.0}$ | $\mathbf{86.1}_{\pm0.4}$ | $\mathbf{89.4}_{\pm0.3}$ | $\mathbf{66.7}_{\pm0.6}$ | $40.6_{\pm0.6}$ | $\mathbf{42.8}_{\pm0.3}$ | $98.5_{\pm0.2}$ |

## 6.2 Math and Coding Tasks

Tables 4b, 4c shows that ALT substantially boosted the LLaMA-3.1-70B's performance by up to 7 and 10 points on math and coding tasks, respectively. The math improvements are reasonable, as understanding predicate logic is a prerequisite for solving mathematical problems. For coding, some recent studies have verified the opposite direction, namely, that training on coding data improves logical reasoning abilities (Jiang et al., 2024b; MA et al., 2024; Uchiyama et al., 2024).

## 6.3 NLI Tasks

Table 4d shows that ALT substantially boosted the LLaMA-3.1-70B's performance by up to 6 points on various natural language inference (NLI) benchmarks. NLI is similar to deductive reasoning in assessing whether a premise supports or contradicts a hypothesis. However, the main difference is that this judgment requires a rich set of commonsense knowledge beyond the given premise.

Consider the fifth problem in Table 5: by supplementing the given fact "An Indian woman is dancing with her partner" with the commonsense knowledge "If someone is dancing, then he/she is moving.", we can derive the hypothesis "A woman is moving." The sixth problem is more challenging as we have to trace multiple logical steps while supplementing with sufficient commonsense knowledge as follows: "a church choir sings at a church," "baseball is often played at a baseball field," "a person cannot be in two or more places at the same time," "therefore, a church choir cannot sing for baseball."

Since synthetic logic corpora only contain unknown facts, LLMs cannot acquire new knowledge from them. Therefore, the commonsense knowledge used to solve the above problems must have been acquired by the LLMs from pre-training. This suggests that LLMs can integrate their original knowledge with the logical reasoning capabilities newly acquired from ALT to solve problems.

Table 5: Problems where LLaMA-3.1-70B initially answered incorrectly and then correctly after training with ALT on $FLD_{\times 2}$. Red highlights the premises related to the hypothesis.

| benchmark | premises | hypothesis | answer (LLaMA-3.1-70B/gold) | required ability |
|---|---|---|---|---|
| LogicNLI | Mice are afraid of wolves. Cats are afraid of sheep. Jessica is a cat. Wolves are afraid of cats. Winona is a wolf. Sheep are afraid of cats. | Jessica is afraid of sheep. | neutral/ entailment | DP1 |
| | Rhett is not modest. Vivian is confused. Rhett is lazy. If someone is modest or not confused, then he is not eager. | Rhett is confused. | entailment/ neutral | DP2 |
| FOLIO | All eels are fish. No fish are plants. Everything displayed in the collection is either a plant or an animal. All animals displayed in the collection are multicellular. A sea eel is displayed in the collection. The sea eel is an eel or an animal or not a plant. | The sea eel is multicellular or is bacteria. | neutral/ entailment | DP3 |
| | Common utilities include water, electricity, gas, heating, sewer, trash, and recycling. Many apartment rents cover the cost of water and electricity. Susan lives in an apartment where the rent covers all utilities. The rent of the apartment where Ava lives does not cover any utility expenses. Noah lives in an apartment where the rent does not cover heating. | Noah and Ava both need to pay the heating bill. | neutral/ entailment | DP4 |
| SNLI | An Indian woman is dancing with her partner. | A woman is moving. | neutral/ entailment | reasoning with commonsense knowledge |
| | This church choir sings to the masses as they sing joyous songs from the book at a church. | A choir is singing at a baseball game. | entailment/ contradiction | |

Table 6: Problems that LLaMA-3.1-70B trained with ALT on $FLD_{\times 2}$ still cannot solve.

| benchmark | question | answer |
|---|---|---|
| ARC (challenge) | The end result in the process of photosynthesis is the production of sugar and oxygen. Which step signals the beginning of photosynthesis? | Chlorophyll in the leaf captures light energy. |
| GPQA | A spin-half particle is in a linear superposition $0.8|\uparrow\rangle + 0.6|\downarrow\rangle$ of its spin-up and spin-down states. If $|\uparrow\rangle$ and $|\downarrow\rangle$ are the eigenstates of $\sigma_z$, then what is the expectation value up to one decimal place, of the operator $10\sigma_z + 5\sigma_x$? | $-0.7$ |
| ARC (challenge) | Beavers build their homes in ponds and streams. Which characteristic is least critical to building homes in an aquatic environment? | (A) waterproof fur (B) webbed hind feet (C) arge, sharp teeth (D) flat, wide tail |

## 6.4 Other Tasks

Improvements across various other tasks (Table 4e) demonstrate the broad benefits of the obtained reasoning capabilities beyond standard reasoning tasks; though the improvements were modest at up to 2 percentage points, which may be due to the following reasons. First, these benchmarks include problems that purely test knowledge, such as the first one in Table 6. Since ALT does not aim to provide new knowledge, the ability to solve such problems does not improve by nature. Next, some problems may require knowledge that is too advanced for LLMs, so potential improvements by the enhanced reasoning capabilities may be bottlenecked. For example, the second problem does involve reasoning but requires sufficient quantum mechanics knowledge as a prerequisite. However, these knowledge-related issues should be solved by improving the quantity and quality of pre-training.

Finally, LLMs may not be able to fully utilize the potential of enhanced reasoning capabilities for problems that require complex procedures. To solve the third problem, LLMs first must attempt reasoning related to each choice as follows: "To build homes in an aquatic environment, one needs to maintain body heat and insulation despite being frequently submerged in cold water. Therefore, the waterproof fur of (A) is essential", and "To build . . . , one must gather and process natural materials like wood. Large, sharp teeth of (C) are critical as they allow beavers to cut down trees and shape branches." Next, while reasoning traces on (A) to (D) all seem reasonable, LLMs must choose the single best answer, considering the subtle nuance of the question context, as follows: "Since the question emphasizes the aquatic environment, the least related reasoning trace should be (C)." This complex procedure contrasts with logical reasoning and NLI problems, where LLMs can directly obtain an answer from a single reasoning trace. Previous studies also observed that such procedure on multiple-choice QA problems are challenging for LLMs (Robinson and Wingate, 2023; Zheng et al., 2024; Wang et al., 2024a). Since ALT alone does not teach LLMs such task-specific procedures, additional training on these procedures should be necessary to solve these problems.

## 7 Conclusion

Towards versatile artificial intelligence with reasoning capabilities, we proposed **A**dditional *Logic* **T**raining on synthetic logic samples. We established systematic design principles well-grounded on symbolic logic theory and previous empirical findings. We constructed a corpus named Formal Logic Deduction Diverse ($FLD_{\times 2}$) based on the design principles. We empirically showed that ALT on $FLD_{\times 2}$ substantially enhances the capabilities of state-of-the-art LLMs.

## Acknowledgement

Computational resources of AI Bridging Cloud Infrastructure (ABCI) provided by the National Institute of Advanced Industrial Science and Technology (AIST) were used. We thank Dr. Masaaki Shimizu at Hitachi for the convenience of additional computational resources. We thank Dr. Naoaki Okazaki, a professor at the Tokyo Institute of Technology, for the keen comments.

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

# A Related Work

## A.1 Investigation of Reasoning Capabilities of LLMs

Many studies examine LLMs' reasoning capabilities (Askell, 2020; Rae et al., 2021; Razeghi et al., 2022; Liu et al., 2023b; Turpin et al., 2023; Lanham et al., 2023; Wu et al., 2023; Hodel and West, 2023; Dziri et al., 2023; Dasgupta et al., 2023). Patel et al. (2024) observed LLMs' performance significantly declines as reasoning steps increase in multi-step logical reasoning tasks. Dougrez-Lewis et al. (2024) revealed ChatGPT struggles with abductive reasoning when verifying claims by decomposing their evidence into atomic reasoning steps. Wang et al. (2024b) found that GPT-series models showed significant gaps compared to humans in dealing with inference rules. Parmar et al. (2024) introduced LogicBench and showed that existing LLMs struggle with instances involving complex reasoning and negations. Wan et al. (2024) introduced LogicAsker, which assesses whether LLMs can employ a set of atomic reasoning skills grounded in propositional and predicate logic and found significant gaps in LLMs' learning of logical rules. Bhuiya et al. (2024) proposed a challenging multi-hop reasoning benchmark with seemingly plausible but incorrect multi-hop reasoning chains and found that state-of-the-art LLMs' capabilities to perform multi-hop reasoning is affected by such chains. Mondorf and Plank (2024) introduced TruthQuest, which assesses LLMs' capabilities to conduct suppositional reasoning, i.e., reasoning where each statement can be false, and found that LLMs exhibit significant difficulties solving these tasks. Sprague et al. (2024) introduced a complex multi-step reasoning benchmark, MuSR, and characterized the gaps that remain for techniques like chain-of-thought to perform robust reasoning.

**Biases and Errors**    Ando et al. (2023); Ozeki et al. (2024); Bertolazzi et al. (2024); Eisape et al. (2024) found that LLMs exhibit human-like reasoning biases in syllogistic arguments. Jiang et al. (2024a) found that LLMs exibit "token-biases" in solving logical reasoning problems. Aoki et al. (2024) revealed that LMs rely heavily on heuristics, such as lexical overlap, in the earlier stages of reasoning. Zhao et al. (2024a) constructed a MATHTRAP with carefully designed logical traps into the problem descriptions of MATH and GSM8k and found that while LLMs possess the knowledge required to solve these traps, they do not spontaneously use such knowledge them to handle the problems. Han et al. (2024) found that LLMs exhibit A-Not-B errors similar to human infants, failing to suppress the previously established response pattern during ICL. Liu et al. (2024) found that LLMs often contradict themselves in reasoning tasks involving contextual information understanding or commonsense. Zhou et al. (2024b) found that subtle alterations in the surface form can significantly impact the answer distribution, suggesting that LLMs solve reasoning problems using surface cues. Chen et al. (2024) found that the reasoning performance of LLMs is affected by the order of the premises. Hong et al. (2024); Huang et al. (2024) found that LLMs struggle to identify fallacious reasoning steps accurately, suggesting challenges in self-verification methods.

**Reasoning in Unknown Situation**    Zhao et al. (2024b) found that LLMs struggle with reasoning in uncommon situations. Zhu et al. (2024) introduced a framework to dynamically generate reasoning samples, and LLMs perform worse in those samples. Hu et al. (2024) found that while LLMs can conduct reasoning when relevant knowledge is given in context, they are not proficient at reasoning with knowledge embedded in the training data.

## A.2 Synthetic Logic Corpus for Training LLMs

RuleTaker Clark et al. (2021) proposed a deduction corpus composed of synthetically generated multistep deductive proofs written in natural languages. Each deductive proof (dis-)proves a hypothesis by applying deduction rules multiple times to a given set of facts. They showed that Transformer Vaswani et al. (2017) LMs can solve these problems in the sense that they can predict the final answer (i.e., "proved", "disproved", or "unknown") of each deductive proof given the fact set. Later studies Saha et al. (2020); Dalvi et al. (2021); Tafjord et al. (2021); Sanyal et al. (2022b) showed that generative LMs can generate even the intermediate proofs as well as the final answer. Later studies (Saha et al., 2020; Dalvi et al., 2021; Tafjord et al., 2021; Sanyal et al., 2022b) showed that T5 can generate even the intermediate logical steps as well as the final answer.

PARARULE-Plus (Bao et al., 2022) is the enhanced version of PARARULE (Clark et al., 2021), a variation of RuleTaker, that includes more samples and more logical steps. RoBERTa (Liu et al., 2019) trained on PARARULE-Plus outperformed the models trained on RuleTaker.

Artificial Argument Corpus (Betz et al., 2021) includes single-step deductive reasoning samples constructed from hand-selected deduction rules useful for critical thinking. They showed that the GPT-2 (Radford et al., 2019) trained on this corpus can generalize to solve NLI tasks. However, at the same time, they found that the LM does not generalize well to solve more challenging reasoning tasks such as ARC (Habernal et al., 2018) and LogiQA (Liu et al., 2020).

FLD by Morishita et al. (2023, 2024) is the first synthetic logic corpus based on formal logic theory. It includes multistep deductive reasoning samples constructed from the axioms of first-order predicate logic, which can express any deduction rule due to the completeness theorem. Due to this nature, T5 trained on FLD generalizes most effectively to other synthetic logic corpora, compared to models trained on other corpora.

Gontier et al. (2020) investigated the deductive reasoning capabilities of LMs on a corpus composed of a specific type of multistep inference, kinship relationships on synthetic kinship graphs. They found that LMs can solve this task when there are relatively few proof steps, but it is difficult for them to generalize to solve proof steps longer than those shown in training data. Bostrom et al. (2021) studied how to create realistic natural language expressions that represent deduction rules. To this end, they scraped sentences from Wikipedia using a template-based method and paraphrased them. They showed that training on this corpus helps solve real-world deductive reasoning problems such as EntailmentBank (Dalvi et al., 2021). Pi et al. (2022) used synthetic data from program executors, most notably SQL programs. They verified that this data can enhance numerical reasoning, logical reasoning, and multi-hop reasoning abilities. Trinh et al. (2024) generated 100 million geometry problems and verified that the capability of artificial intelligence can be enhanced to to pass the bronze medal threshold of the International Mathematics Olympiad. Saeed et al. (2021); Nafar et al. (2024) created *soft* reasoning rules involving with probabilistic logic, instead of hard-logic examined by the aforementioned studies. Sileo (2024) introduced a simpler and more general declarative framework for synthetic generation, and verified its effectiveness. Zhou et al. (2024a) synthetically generated a large dataset of mathematics, and gained over 12 points on GSM8k.

While these studies partly examined the effect of synthetic logic corpora, whether this approach is promising remains an open question. It has been unexplored whether the capabilities obtained from synthetic logic corpora generalizes to solve various tasks beyond the original tasks in these corpora. Additionally, the effect of these corpora has only been examined for small LMs trained on small pre-training corpora such as T5 and RoBERTa; it has been highly questionable whether they can still benefit state-of-the-art LLMs trained on a huge pre-training corpus. Furthermore, even if their benefits were verified, it remains unclear which design of synthetic logic samples yields the largest benefits due to the lack of systematic discussions on sample designs and empirical verification of these designs. We aimed to answer these questions in this paper and demonstrate the potential of synthetic logic corpora.

### A.3 Distilling Reasoning Traces from Very Large LLMs

Recent approaches (Ho et al., 2023; Magister et al., 2023; Li et al., 2022, 2023; Shridhar et al., 2023; Wang et al., 2023; Mitra et al., 2023; Liu et al., 2023c; Ben Allal et al., 2024; Lu et al., 2024) utilize very large LLMs, such as GPT-4, to prepare synthetic reasoning datasets to train smaller LLMs. A typical procedure is as follows: (i) prepare existing reasoning problems, (ii) prompt large LLMs to generate reasoning traces to solve these problems using techniques such as chain-of-thought prompting (Wei et al., 2022), and (iii) train smaller LLMs on these reasoning traces.

The distillation approach and the synthetic logic corpora approach examined in this paper have specific advantages and disadvantages, as follows.

The advantage of the distillation approach is its immediate practical effect, as it directly teaches LLMs solutions to various existing problems. The disadvantages could be that (i) it is non-trivial for specific solutions to specific problems to generalize to other problems, (ii) the number of training samples is limited to existing problems in nature, (iii) the correctness and faithfulness of the reasoning traces are not guaranteed; indeed, some studies (Turpin et al., 2023; Lanham et al., 2023) suggest that large LLMs do not always faithfully follow the "reasoning traces" they themselves generate, and (iv) it cannot enhance the very large LLMs themselves by nature.

The advantages of synthetic logic corpus approaches are that (i) since they teach the fundamentals of reasoning, such as deductive reasoning, they have the potential to generalize to various problems,

(ii) they can generate an unlimited number of new samples, and (iii) the correctness of the reasoning traces is guaranteed by nature. The disadvantage of this approach is that, as it only teaches the basics of reasoning, additional training may be needed to solve more complex real-world problems, as suggested in Section 6.4.

We hypothesize that integrating both approaches could be promising. That is, we first train LLMs using ALT to make them understand the fundamentals of reasoning through high-quality samples and then train them using more realistic reasoning traces to solve complex real-world problems.

## B  Limitations

- We only used deductive reasoning samples for ALT. Future work should examine other reasoning samples, e.g., abductive and inductive reasoning.
- We only examined the first-order predicate logic system. Future work should examine other logic systems, such as modal and linear logic.

## C  Ethics and Social Impacts

The ultimate goal of the direction of this study is to develop an AI capable of reasoning logically step by step. If AI can make a decision one logical step at a time, it would be highly explainable and transparent to users. Furthermore, the user would be able to trace the AI's errors. We believe that our study is a step towards such AI that will positively impact society.

## D  Details of Formal Logic Deduction Diverse

Figure D.3 shows a real sample from $FLD_{\times 2}$. Below, We briefly explain our sample generator. Please refer to Morishita et al. (2023) for the details.

### D.1  Answer Labels

In addition to the logical steps, the samples of $FLD_{\times 2}$ and previous corpora include *answer labels* (Figure D.3): "proved" indicating that the hypothesis can be proved by the logical steps, "disproved" indicating that the hypothesis can be disproved, and "unknown" indicating that the given facts are insufficient for either proving or disproving the hypothesis. For samples with "unknown" labels, the logical steps are "None.". $FLD_{\times 2}$ have a uniform distribution over the labels.

### D.2  Splits

$FLD_{\times 2}$ includes 100k/5k/5k samples for train/valid/test splits.

### D.3  Generation of Multistep Deduction

Our sample generator first randomly generates examples of multistep deduction by forward- and backward random deduction, using the deduction rules specified by a user.

The forward random deduction is done as follows. The generator first chooses a deduction rule randomly and forms the initial tree where the root node is the conclusion of the chosen deduction rules and the child nodes are the premises of the chosen deduction rule. The generator next randomly chooses another deduction rule that can be "jointed" to the root note of the tree. A deduction rule can be jointed to the root node of a tree if one of the premises of that deduction rule can be identified with the root node. Then, the generator updates the tree by jointing this chosen deduction rule. The generator continues this step multiple times until the tree achieves the required depth.

The backward random deduction is done as follows. For each step, the generator randomly chooses a leaf node of the tree. Then, the generator randomly chooses a deduction rule that can be jointed to the leaf node. Here, a deduction rule can be jointed to the leaf node if the deduction rule's conclusion can be identified with the leaf node. Then, the generator updates the tree by jointing this chosen deduction rule. The generator continues this step multiple times until the complexity of branches achieves the required level.

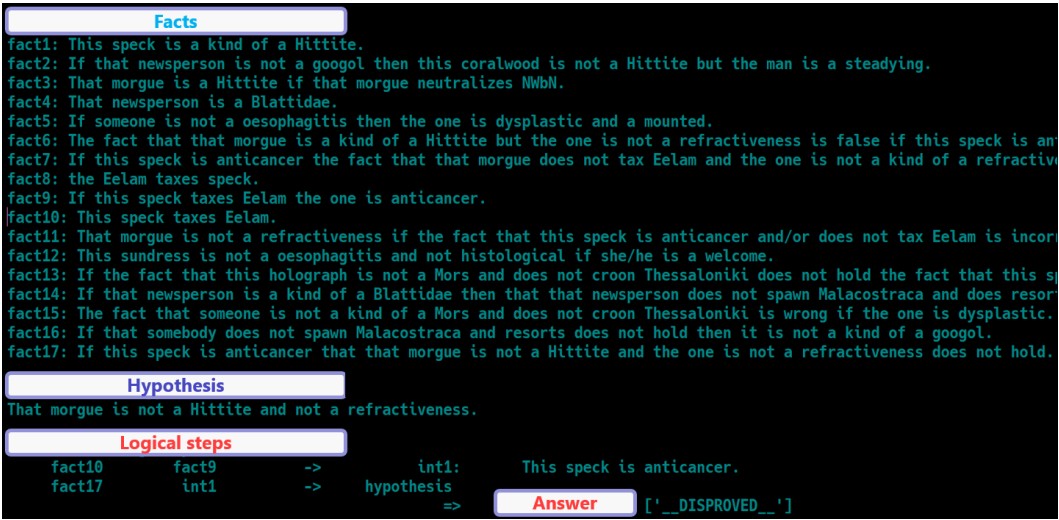

Figure D.3: A real deduction sample included in Formal Logic Deduction Diverse. **Facts** and **hypothesis** are given to LLMs, then the LLMs are required to generate **logical steps** to (dis-)prove the hypothesis based on the facts, and an **answer** label (see Appendix D.2).

## D.4 Linguistic Expressions

We prepared linguistic templates for each logical formula, exemplified as follows:

$$
\begin{aligned}
\langle (A \wedge B) \rightarrow C \rangle &: \text{If } \langle (A \wedge B).\text{predicate\_phrase} \rangle, \text{ then } \langle C.\text{predicate\_phrase} \rangle. \\
&: \langle (A \wedge B).\text{noun\_pharse} \rangle \ \langle \text{cause\_synonyms} \rangle \ \langle C.\text{noun\_phrase} \rangle. \\
&: (\dots) \\
\langle (A \wedge B).\text{predicate\_phrase} \rangle &: A \ \langle \text{occur\_synonyms} \rangle \text{ and also } B \ \langle \text{occur\_synonyms} \rangle. \\
&: A \text{ and also } B \ \langle \text{occur\_synonyms} \rangle. \\
&: \text{Both } A \text{ and } B \ \langle \text{occur\_synonyms} \rangle. \\
&: (\dots) \\
\langle C.\text{predicate\_phrase} \rangle &: C \ \langle \text{occur\_synonyms} \rangle. \\
&: (\dots) \\
\langle \text{occur\_synonyms} \rangle &: \text{occur} \\
&: \text{happen} \\
&: \text{take place} \\
&: (\dots) \\
\langle (A \wedge B).\text{noun\_pharse} \rangle &: A \text{ and } B \\
&: A \text{ and also } B \\
&: \text{Both } A \text{ and } B \\
&: \text{That } A \text{ and } B \ \langle \text{occur\_synonyms} \rangle \\
&: (\dots) \\
\langle \text{cause\_synonyms} \rangle &: \text{cause} \\
&: \text{result in} \\
&: \text{lead to} \\
&: \text{bring about} \\
&: (\dots) \\
(\dots) & \tag{D.1}
\end{aligned}
$$

As can be seen, the templates can be nested deeply, yielding combinatorially diverse linguistic expressions.

Expanding these templates beforehand is intractable due to the combinatorial explosion, so we expand these templates on the fly to randomly sample a single expression at a time. Estimating the exact number of expressions is intractable for the same reason.

We manually crafted several additional English templates per logical formula (i.e., the left-hand sides of (D.1)) compared to those used in FLD, which yield combinatorially more diverse English

expressions. We observed that at least dozens of expressions, including minor variations, are yielded for each formula.

# E   Details of Experimental Setup

### E.0.1   Prevention of Knowledge Forgetting by Recall Adam Optimizer

We employed the Recall Adam (RecAdam) optimizer (Chen et al., 2020), which regularizes parameter updates to prevent them from being too far from the pre-training parameters. Recall Adam stands out for LLM training as it does not require access to the pre-training corpus, which is often inaccessible or too huge to handle, nor does it require changes to the model architecture, and it has a proven track record of usage in language models such as BERT.

## E.1   Benchmarks

Table E.7 details the benchmarks used in the experiments.

## E.2   Experimental Runs

We show the average and standard deviations over five seeds.

## E.3   Computational Resources

The entire experiment, including preliminary ones, took about 1 week x 128 NVIDIA H100 GPUs of our own.

# F   Results without using Recall Adam

Table F.8 shows the results of LLMs trained without using Recall Adam.

Table E.7: 31 benchmarks used in the experiments. These benchmarks cover a wide range of tasks and are prominent for LLM evaluation. We also show the form of reasoning and the type of knowledge required to solve the problems in each benchmark.

| Set | Benchmarks | Reasoning form | Required knowledge |
|---|---|---|---|
| Logic | bAbi deduction (Weston et al., 2015), FOLIO (Han et al., 2022) LogicNLI (Tian et al., 2021) RobustLR (Sanyal et al., 2022a) | deduction | - (not required) |
| | AR-LSAT (Zhong et al., 2021) LogiQA2 (Liu et al., 2023a) ReClor (Yu et al., 2020) | | commonsense |
| | AbductionRules (Young et al., 2022) | abduction | |
| | ART (Bhagavatula et al., 2019) | | commonsense |
| NLI | HELP (Yanaka et al., 2019) MultiNLI (Williams et al., 2018) RTE (Dagan et al., 2005; Giampiccolo et al., 2007; Bentivogli et al., 2009) SNLI (Bowman et al.) | validate a conclusion based on given premises | commonsense |
| Math | GSM8k (Cobbe et al., 2021) MATH (Hendrycks et al., 2021b) MathQA (Amini et al., 2019) | Math | Math |
| Coding | HumanEval (Chen et al., 2021) MBPP (Austin et al., 2021) MBPP+ (Liu et al., 2023d) MultiPL-E (cpp/go) (Cassano et al., 2023) | Coding | Coding |
| Others | CommonsenseQA (Talmor et al., 2018) HellaSWAG (Zellers et al., 2019) SQuAD2 (Rajpurkar et al., 2018) WinoGrande (Sakaguchi et al., 2021) | complicated procedures | commonsense |
| | ARC (easy/challenge) (Clark et al., 2018) GPQA (Rein et al., 2023) OpenBookQA (Mihaylov et al., 2018) SciQ (Welbl et al., 2017) | | science |
| aggregated | MMLU (Hendrycks et al., 2021a) MMLU-Pro (Wang et al., 2024c) BBH (Suzgun et al., 2022) | various | various |

Table F.8: 5-shot performance of LLMs before and after ALT. ⊕**ALT**-$x$ denotes the LLM trained with ALT on the synthetic logic corpus $x$ from Table 1. Color shows the rank in each column (darker is better). "Logic", "Math", "Code", and "Others" each comprises various benchmarks (see Table E.7). "Avg." represents the micro-average of all the benchmarks. "w/o RecAdam" denotes that LLM was trained without knowledge forgetting prevention by Recall Adam optimizer.

(a) LLaMA-3.1-8B.

| | Avg. | Logic | Math | Code | NLI | Others | BBH (3-shot) | | BBH (0-shot) | | MMLU | |
|---|---|---|---|---|---|---|---|---|---|---|---|---|
| | | | | | | | | CoT | | CoT | | Pro |
| LLaMA-3.1-8B | 47.9 | $42.8_{\pm0.4}$ | $39.6_{\pm0.5}$ | 35.4 | $65.4_{\pm0.3}$ | $60.7_{\pm0.3}$ | $44.9_{\pm0.2}$ | $61.9_{\pm0.4}$ | $8.2_{\pm0.2}$ | $36.5_{\pm0.4}$ | $65.3_{\pm0.4}$ | $35.8_{\pm0.4}$ |
| ⊕ALT-PRP $_{\text{w/o RecAdam}}$ | 43.5 | $39.5_{\pm0.2}$ | $29.1_{\pm0.3}$ | 35.3 | $57.8_{\pm0.2}$ | $61.0_{\pm0.2}$ | $40.5_{\pm0.2}$ | $47.0_{\pm0.2}$ | $3.9_{\pm0.1}$ | $6.3_{\pm0.1}$ | $64.9_{\pm0.2}$ | $34.0_{\pm0.2}$ |
| ⊕ALT-PRP | 48.1 | $43.7_{\pm0.2}$ | $39.2_{\pm0.3}$ | 35.7 | $65.6_{\pm0.2}$ | $60.8_{\pm0.2}$ | $44.9_{\pm0.2}$ | $61.8_{\pm0.2}$ | $8.2_{\pm0.1}$ | $36.4_{\pm0.2}$ | $65.3_{\pm0.2}$ | $35.3_{\pm0.2}$ |
| ⊕ALT-RT | 50.1 | $46.8_{\pm0.1}$ | $42.4_{\pm0.2}$ | 36.5 | $68.6_{\pm0.1}$ | $61.3_{\pm0.1}$ | $46.9_{\pm0.2}$ | $63.5_{\pm0.2}$ | $13.7_{\pm0.1}$ | $38.4_{\pm0.2}$ | $65.3_{\pm0.1}$ | $35.7_{\pm0.2}$ |
| ⊕ALT-FLD | 51.9 | $51.6_{\pm0.1}$ | $43.4_{\pm0.1}$ | 38.1 | $70.1_{\pm0.1}$ | $61.5_{\pm0.1}$ | $46.7_{\pm0.2}$ | $64.9_{\pm0.2}$ | $11.9_{\pm0.1}$ | $39.6_{\pm0.2}$ | $65.4_{\pm0.1}$ | $36.2_{\pm0.2}$ |
| ⊕**ALT-FLD**$_{\times2}$ | 52.0 | $52.2_{\pm0.1}$ | $43.2_{\pm0.2}$ | 38.0 | $70.7_{\pm0.1}$ | $61.5_{\pm0.1}$ | $46.5_{\pm0.2}$ | $65.3_{\pm0.2}$ | $11.3_{\pm0.1}$ | $38.7_{\pm0.2}$ | $65.5_{\pm0.1}$ | $36.4_{\pm0.2}$ |

(b) LLaMA-3.1-70B.

| | Avg. | Logic | Math | Code | NLI | Others | BBH (3-shot) | | BBH (0-shot) | | MMLU | |
|---|---|---|---|---|---|---|---|---|---|---|---|---|
| | | | | | | | | CoT | | CoT | | Pro |
| LLaMA-3.1-70B | 60.0 | $57.4_{\pm0.4}$ | $60.0_{\pm0.5}$ | 46.2 | $73.7_{\pm0.3}$ | $67.7_{\pm0.3}$ | $60.4_{\pm0.3}$ | $82.1_{\pm0.2}$ | $6.5_{\pm0.1}$ | $50.1_{\pm0.3}$ | $78.7_{\pm0.3}$ | $50.7_{\pm0.4}$ |
| ⊕ALT-PRP $_{\text{w/o RecAdam}}$ | 58.8 | $54.3_{\pm0.4}$ | $59.2_{\pm0.5}$ | 48.2 | $72.7_{\pm0.3}$ | $65.9_{\pm0.3}$ | $60.4_{\pm0.4}$ | $81.5_{\pm0.3}$ | $6.1_{\pm0.2}$ | $48.3_{\pm0.4}$ | $78.5_{\pm0.3}$ | $50.7_{\pm0.4}$ |
| ⊕ALT-PRP | 60.4 | $57.7_{\pm0.4}$ | $59.8_{\pm0.5}$ | 49.2 | $73.5_{\pm0.3}$ | $67.6_{\pm0.3}$ | $60.4_{\pm0.4}$ | $82.2_{\pm0.3}$ | $6.0_{\pm0.2}$ | $50.1_{\pm0.4}$ | $78.7_{\pm0.3}$ | $50.9_{\pm0.4}$ |
| ⊕ALT-RT | 62.7 | $61.4_{\pm0.2}$ | $62.1_{\pm0.3}$ | 50.8 | $75.4_{\pm0.2}$ | $68.4_{\pm0.2}$ | $64.1_{\pm0.3}$ | $82.5_{\pm0.2}$ | $11.5_{\pm0.2}$ | $59.2_{\pm0.3}$ | $79.0_{\pm0.2}$ | $52.4_{\pm0.3}$ |
| ⊕ALT-FLD | 64.2 | $65.7_{\pm0.1}$ | $63.6_{\pm0.2}$ | 52.0 | $75.3_{\pm0.1}$ | $68.5_{\pm0.1}$ | $65.0_{\pm0.2}$ | $83.6_{\pm0.1}$ | $12.1_{\pm0.1}$ | $59.9_{\pm0.2}$ | $79.3_{\pm0.1}$ | $54.4_{\pm0.2}$ |
| ⊕**ALT-FLD**$_{\times2}$ | 64.4 | $66.1_{\pm0.1}$ | $63.3_{\pm0.2}$ | 52.4 | $76.1_{\pm0.1}$ | $68.5_{\pm0.1}$ | $65.4_{\pm0.2}$ | $83.6_{\pm0.2}$ | $11.4_{\pm0.1}$ | $60.8_{\pm0.2}$ | $79.5_{\pm0.1}$ | $54.4_{\pm0.2}$ |

Table F.9: Benchmark-wise 5-shot performance of LLaMA-3.1-8B before and **after** ALT on FLD$_{\times2}$.

(a) Logic.

| | bAbiD | FOLIO | LogicNLI | RobustLR | AR-LSAT | LogiQA | ReClor | AbductionR | ART |
|---|---|---|---|---|---|---|---|---|---|
| LLaMA-3.1-8B | $48.7_{\pm1.6}$ | $50.0_{\pm1.6}$ | $28.5_{\pm1.0}$ | $43.2_{\pm0.9}$ | $20.7_{\pm1.0}$ | $39.6_{\pm1.2}$ | $28.7_{\pm0.7}$ | $52.4_{\pm0.9}$ | $73.4_{\pm1.1}$ |
| ⊕**ALT-FLD**$_{\times2}$ | $\mathbf{55.8}_{\pm0.6}$ | $\mathbf{54.5}_{\pm0.6}$ | $\mathbf{42.0}_{\pm0.4}$ | $\mathbf{62.6}_{\pm0.3}$ | $\mathbf{21.1}_{\pm0.4}$ | $\mathbf{42.8}_{\pm0.4}$ | $\mathbf{29.4}_{\pm0.2}$ | $\mathbf{85.5}_{\pm0.2}$ | $\mathbf{76.1}_{\pm0.4}$ |

(b) Math.

| | GSM8k | | | MATH | MathQA |
|---|---|---|---|---|---|
| | | CoT | CoT (0-shot) | - | - |
| LLaMA-3.1-8B | $50.2_{\pm1.4}$ | $51.5_{\pm1.4}$ | $39.5_{\pm1.3}$ | $14.1_{\pm0.5}$ | $42.8_{\pm0.9}$ |
| ⊕**ALT-FLD**$_{\times2}$ | $\mathbf{53.6}_{\pm0.5}$ | $\mathbf{56.4}_{\pm0.5}$ | $\mathbf{48.4}_{\pm0.5}$ | $\mathbf{14.3}_{\pm0.2}$ | $\mathbf{43.3}_{\pm0.3}$ |

(c) Coding.

| | HumanEval | MBPP | MBPP+ | MultiPL-E (cpp) | MultiPL-E (go) |
|---|---|---|---|---|---|
| LLaMA-3.1-8B | 22.6 | 31.6 | 38.1 | 21.7 | 63.0 |
| ⊕**ALT-FLD**$_{\times2}$ | **25.9** | **34.0** | **39.9** | **23.0** | **67.1** |

(d) Natural language inference (NLI).

| | HELP | MNLI | RTE | SNLI |
|---|---|---|---|---|
| LLaMA-3.1-8B | $46.4_{\pm0.5}$ | $68.1_{\pm0.5}$ | $74.6_{\pm0.9}$ | $72.6_{\pm0.4}$ |
| ⊕**ALT-FLD**$_{\times2}$ | $\mathbf{47.9}_{\pm0.2}$ | $\mathbf{75.3}_{\pm0.2}$ | $\mathbf{83.1}_{\pm0.3}$ | $\mathbf{76.5}_{\pm0.1}$ |

(e) Others.

| | CommonsenseQA | HellaSwag | SQuAD | WinoGrande | ARCe | ARCc | GPQA | OpenBookQA | SciQ |
|---|---|---|---|---|---|---|---|---|---|
| LLaMA-3.1-8B | $73.9_{\pm1.3}$ | $61.2_{\pm0.5}$ | $30.8_{\pm0.0}$ | $77.4_{\pm1.2}$ | $84.2_{\pm0.7}$ | $54.7_{\pm1.5}$ | $\mathbf{31.1}_{\pm1.3}$ | $35.3_{\pm0.7}$ | $\mathbf{97.7}_{\pm0.5}$ |
| ⊕**ALT-FLD**$_{\times2}$ | $\mathbf{74.8}_{\pm0.4}$ | $\mathbf{61.5}_{\pm0.2}$ | $\mathbf{33.5}_{\pm0.0}$ | $\mathbf{78.1}_{\pm0.5}$ | $\mathbf{85.0}_{\pm0.3}$ | $\mathbf{55.6}_{\pm0.5}$ | $31.1_{\pm0.5}$ | $\mathbf{36.3}_{\pm0.2}$ | $97.6_{\pm0.2}$ |

