# OpenReview forum: "Enhancing Reasoning Capabilities of LLMs via Principled Synthetic Logic Corpus"
_NeurIPS.cc/2024/Conference — NeurIPS 2024 poster_

### Official Review · Reviewer_BW1m · 2024-07-09

**Soundness:** 3
**Presentation:** 2
**Contribution:** 2
**Rating:** 4
**Confidence:** 3

**Summary:**

This paper aims to improve LLM's logical reasoning ability by constructing synthetic data used in continual training. This work is largely built upon FLD, and proposes four other design principles for the synthetic dataset. Namely, reasoning with unknown facts, illogical reasoning, diverse reasoning rules, and linguistic expressions.  By training llama-7b and 70b models on the synthetic dataset, the authors proved the effectiveness of their methods by achieving improvement on various benchmarks such as BBH.

**Strengths:**

1. The experiments are pretty solid. The authors tested ALPT on different scales of LLMs on various NLP tasks such as logical reasoning, reading comprehension, etc.
2. According to Table 2, the performance gain on 70b model is even larger than 7b model, which shows the potential of ALPT.
3. The paper is organized well.
4. The authors provide an anonymous link for all the code, model, and data to ensure reproducibility.

**Weaknesses:**

1.  I am mostly concerned about the contribution of this paper. It seems to me that the four design principles are incremental improvements based on [1] by increasing the vocab size and enriching the diversity of logical rules and linguistic expressions. The results in Table 2 seem incremental compared with FLD baselines either. Also, the writing of some parts of the paper is very similar to [1], with even the same examples.
2. The current writing makes it hard for non-experts to understand. Specifically, the description of how the generator works is listed in the appendix. If one hasn't read [1] or the appendix, it would be very hard to understand the overall workflow.
3. It would strengthen the paper if the authors could also provide the results for FLD in Table 3 as a baseline for comparison.


[1] Morishita, Terufumi, et al. "Learning deductive reasoning from synthetic corpus based on formal logic." International Conference on Machine Learning. ICML, 2023.

**Questions:**

1. I think the training process using the constructed synthetic data is more like a continual training or SFT setting. Calling it additional "pre-training" could be a little misleading.

**Limitations:**

There is a limitation section in the appendix.

---

> ### Author Rebuttal · Authors · 2024-08-07
>
> We appreciate your constructive feedback.
> Below, we will address your questions to the best of our ability.
>
> > the training process is more like a continual training
>
> We greatly appreciate your observation.
> Indeed, *pre*-training doesn't make sense much.
> We will rename it to **C**ontinual **L**ogic **T**raining.
>
> > the description of how the generator works is listed in the appendix.
>
> We will add a new subsection in Section 3 to explain the workflow.
> We will include content from the appendix and a new diagram illustrating the workflow.
>
> > It would strengthen the paper if the authors could also provide the results for FLD in Table 3
>
> We will add the task-specific results (i.e., Table 3) of baseline corpora (RT, PRP and FLD) to the main text if space permits; otherwise, we will include them in the appendix.
>
> > I am mostly concerned about the contribution of this paper. It seems to me that the four design principles are incremental improvements based on [1] by increasing the vocab size and enriching the diversity of logical rules and linguistic expressions. The results in Table 2 seem incremental compared with FLD baselines either.
>
> * We first emphasize the higher-level, more fundamental contribution of this study beyond updates from the previous studies: it is the first comprehensive study to verify the promise of synthetic logic corpora to enhance LLMs' reasoning capabilities (detailed below in `D.1`).
> * Regarding the design principles (DPs), while the final suggestions in Table 1 may seem incremental, the comprehensive and systematic analyses of these principles (Sec2 and the experiments) are novel and crucial in the field of synthetic logic corpus, establishing a foundation for future research (detailed below in `D.2`).
> * We consider the Llama-3-70B results in Table 2 (b) significant improvements over the FLD corpus.
>
> # D.1: Our study is the first to verify the promise of synthetic logic corpus through comprehensive experiments.
> RuleTaker (2020,[14]) initially proposed a synthetic logic corpus approach.
> Follow-up studies proposed various corpora, including ProofWriter [63], PARARULE-Plus [3], AACorpus [7], and FLD [51].
> These studies aim to enhance reasoning capabilities of language models with high-quality samples, ultimately achieving versatile AI.
>
> Despite conceptual promise, the effectiveness of the synthetic logic corpus remains uncertain due to lack of comprehensive empirical verification.
> Previous studies examined small language models (SLMs) trained on limited pre-training corpora, such as T5.
> Thus, a fundamental question remains: **RQ1: "Can a synthetic logic corpus enhance reasoning capabilities of state-of-the-art large language models (LLMs) trained on massive pre-training corpora?"**
> This question is non-trivial because massive pre-training corpora might have already taught LLMs reasoning, potentially negating the effect of synthetic logic corpora.
> Moreover, previous studies examined a limited range of evaluation tasks, raising **RQ2: "Can capabilities obtained from synthetic logic corpora generalize to different tasks?"**
> Answering these questions is crucial to advancing the synthetic logic corpus approach further.
>
> Our study addressed these questions through comprehensive experiments, as follows:
> * RQ1: Synthetic logic corpora significantly enhance state-of-the-art LLMs' capabilities, including Llama-3-70B, one of the largest LLMs trained on over 15 trillion tokens (Table 2).
> * RQ2: Capabilities obtained from synthetic logic corpora generalize to various task types (Table 3, and Table F.9 in the newly attached PDF on OpenReview, which includes math and coding tasks).
>
> We emphasize that we demonstrated the positive effect of the synthetic logic corpus approach *as a whole*, involving *not only ALPT-PLD but also ALPT-RT and ALPT-FLD*.
>
> In summary, our study's fundamental contribution provides comprehensive verification of the synthetic logic corpus approach.
> This establishes a promising direction for developing versatile AI with reasoning capabilities.
> (We sincerely apologize for including these important discussions only in Appendix A1; we will include them in the main text.)
>
> # D.2: DPs provide the first comprehensive and systematic analyses of synthetic corpus design, accelerating future research in this field.
> In synthetic generation, computer programs create samples based on pre-designed patterns, which significantly influence sample quality.
> Previous studies developed several corpora with different designs [14,7,71], but lack comprehensive and systematic analyses on these designs.
> Important questions remain, such as **"What aspects are crucial for the design?"** and **"*Why* are these aspects important?"**
>
> We addressed these questions by enumerating crucial design aspects as principles and discussing why each principle is important, considering symbolic logic theory and integrating empirical insights from previous studies.
> For example:
> * A large vocabulary is important *because* it teaches reasoning with unknown facts (DP1).
> * Hard distractors are important *because* they teach when *not* to derive conclusions (DP2).
> * For reasoning rules, while axioms are theoretically sufficient [23,51], theorems are also important *due to* LLMs' limited capability to handle long logical proofs, which was observed in several studies [24] (DP3).
> * We need to include diverse linguistic expressions for logical formulas *because* otherwise, LLMs could overfit to specific linguistic expressions in the corpus, which was observed in several studies [90,92] (DP4).
>
> We also verified the importance of each principle through ablation experiments (Figure 2) and analyzed how these principles influence LLMs' output generations through case studies (Table 4).
>
> Our comprehensive and systematic analyses are novel in the field of synthetic logic corpus.
> These analyses offer insights into future research in this field. For example, researchers can apply, critique, or update the proposed principles.

---

### Official Review · Reviewer_zCCP · 2024-07-09

**Soundness:** 3
**Presentation:** 4
**Contribution:** 3
**Rating:** 7
**Confidence:** 4

**Summary:**

This work proposes Additional Logic Pre-Training (ALPT) to enhance logical reasoning abilities using synthetic rule-based data. The paper first discusses the design principles for creating a logical corpus and subsequently builds PureLogicDiverse (PLD). By training on PLD with RecAdam, models demonstrate improved performance in logical reasoning across various tasks and datasets, effectively integrating enhanced logical reasoning capabilities with their inherent knowledge. The results confirm the effectiveness of the proposed dataset and the ALPT strategy.

**Strengths:**

1. The paper is well-written and presents a fluent narrative from the design principles to the creation of the dataset.
2. Compared to previous logical datasets, PureLogicDiverse contains more comprehensive and complex scenarios, serving as a useful resource for the community.
3. After additional logical pre-training on PLD, different models achieve better performance on both logical reasoning and NLI tasks, with no loss on other tasks. This training strategy together with the PLD dataset can be of interest for the development of future models.
4. The experiments are solid and sound. The comparisons and ablations are comprehensive, providing clear analyses of the effects of different design principles.

**Weaknesses:**

The overall experiments are well-established. However, when comparing the ablation results with those of previous work (Table 2a and Figure 2), RuleTakers appears to perform well enough given the simplicity of its design. For example, comparing ‘ALPT-RT’(Table 2a) with ‘w/o DP2’(Figure 2), despite ‘w/o DP2’ containing much bigger vocabulary size and more extensive rules and expressions, these two settings gain similar performances.

To clarify these comparisons, it would be helpful if the authors could provide statistics on the different dataset settings, such as the number of steps contained in the sampled training splits of each corpus and the average number of rules/steps per sample.

**Questions:**

1. In dataset creation (Line 188), when sampling from the predefined logical formulas, is there a prior distribution from which to sample, or are the formulas evenly distributed? Do you have statistics on the operators and formulas of the created dataset?
2. Since the nouns and predicates are randomly composed to form logical expressions, they lack practical meanings. Intuitively, the coverage of rules and formulas plays a more important role than vocabulary coverage. Considering this, is covering a wide range of 15k vocabulary really necessary? As shown in Figure 2 (w/o DP1), restricting the vocabulary size appears to have minimal impact on the final performance and could also help decrease the dataset size, potentially improving data efficiency. Do you have any insights on this?

**Limitations:**

The authors have reasonably discussed the limitations and potential societal impacts of their research.

---

> ### Author Rebuttal · Authors · 2024-08-07
>
> We appreciate your constructive feedback.
> Below, we will address your questions to the best of our ability.
>
>
> > (...) restricting the vocabulary size appears to have minimal impact on the final performance and could also help decrease the dataset size, potentially improving data efficiency. Do you have any insights on this?
>
> We agree on the importance of dataset efficiency.
> We conducted additional experiments to investigate whether restricting vocabulary (i.e., w/o DP1) has minimal impact, and whether we can reduce the dataset size.
> Table F.9 (in the PDF newly attached to OpenReview) shows that:
> 1. w/o DP1 significantly degraded the performance of LLMs other than Llama3-8B, which was originally shown in the paper. This indicates that vocabulary size is important.
> 1. Reducing the dataset size also degraded performance significantly. This suggests a trade-off between dataset size and final performance, making it challenging to reduce dataset size without performance degradation.
>
> While we maintain the full-vocabulary version as the official corpus, we will allow users to choose their preferred trade-off by using our generator's "--limit-vocab-size" option (https://anonymous.4open.science/r/PLD-generator/scripts/create_corpus.py).
>
> We appreciate the suggestion for new experiments and will include these additional results in the paper's final version.
>
>
>
> > To clarify these comparisons, it would be helpful if the authors could provide statistics on the different dataset settings,
>
> We calculated some statistics for PLD using 100,000 examples:
>
> | total labels                                       | vocab size | total distractors | unique rules | total steps |
> |----------------------------------------------------|------------|-------------------|--------------|-------------|
> | proved(33,391)/ disproved(33,610)/ unknown(32,999) | 100,000    | 1,008,725         | 50           | 187,416     |
>
> We will enumerate all other PLD attributes, such as unique rules and logical operators, and include them in the paper's final version.
> We will attempt to count similar statistics for the baseline corpora.
> Ideally, we will add statistics for all corpora, including PLD and the baselines, to provide a comprehensive comparison.
>
>
>
> > when sampling from the predefined logical formulas, is there a prior distribution from which to sample, or are the formulas evenly distributed?
>
> To avoid overly complex formulas, we use logical formulas with up to three predicates in a sentence, such as "ForAll(x): F(x)", "ForSome(x): (F(x) AND ^G(x)) -> H(x)", and "^(F(a) OR G(b)) -> H(c)", where "^" indicates negation.
> This setting aligns with the previous studies.
> We sample these formulas evenly.

---

> > ### Comment · Reviewer_zCCP · 2024-08-13
> >
> > Thank you for your response. I believe this work is solid as it stands. I will be maintaining my original score.

---

### Official Review · Reviewer_aRM4 · 2024-07-13

**Soundness:** 3
**Presentation:** 3
**Contribution:** 3
**Rating:** 8
**Confidence:** 3

**Summary:**

The paper discusses a novel approach to improve the logical reasoning capabilities of large language models (LLMs). The authors propose a method called Additional Logic Pre-Training (ALPT), which involves training LLMs on a synthetic corpus named PureLogicDiverse. This corpus is designed to include high-quality, program-generated reasoning samples that adhere to strict logical principles.

**Strengths:**

The approach significantly enhance LLMs in logical reasoning abilities. Empirical results show that models pre-trained with ALPT on the PureLogicDiverse corpus perform much better, especially on benchmarks like BBH and NLI tasks. This demonstrates that ALPT can make LLMs more versatile and capable of handling various reasoning tasks.

The systematic design of the PureLogicDiverse corpus ensures that the reasoning samples are high quality, covering a wide range of logical rules and linguistic expressions. This comprehensive method not only boosts logical reasoning but also helps models integrate existing knowledge with new reasoning skills. As a result, they become more effective in tasks requiring both logic and knowledge.

**Weaknesses:**

There is a risk of models overfitting to the synthetic logic patterns, which could limit their generalizability to real-world applications. The success of ALPT heavily depends on the quality of the design principles used to create the synthetic corpus, and any deficiencies in these principles could reduce its effectiveness. Therefore, while ALPT shows great promise, it comes with challenges that need careful management to maximize its potential benefits.

**Questions:**

Authors discussed about the performance on other tasks like CommonsenseQA and Hellaswag, the results show that there is no substantial improvement on these tasks. I am curious to see the performance on other tasks like math or coding.

**Limitations:**

The approach may not fully address tasks requiring complex procedural understanding or multiple-choice questions that involve nuanced reasoning. Further research is needed to integrate ALPT with other methods to enhance performance on these types of tasks.
he effectiveness of ALPT heavily depends on the quality and comprehensiveness of the design principles used to create the synthetic corpus. Any biases or limitations in these principles could affect the overall performance improvements and generalizability of the models.

---

> ### Author Rebuttal · Authors · 2024-08-07
>
> We appreciate your constructive feedback.
> Below, we will address your questions to the best of our ability.
>
> > I am curious to see the performance on other tasks like math or coding.
>
> We conducted additional experiments on math and coding tasks.
> Table F.9 (in the newly attached PDF on OpenReview) shows that **ALPT significantly enhances the LLM's performance on various math and coding benchmarks.**
> These results are very interesting to us; since our samples use first-order predicate logic, which differs much from math and coding, we did not expect improvements in math and coding tasks.
>
> We will include these findings in the paper's final version to further strengthen our argument.
> Thank you very much for the suggestion!
>
>
> > (...) Therefore, while ALPT shows great promise, it comes with challenges that need careful management to maximize its potential benefits.
>
> While our experiments (Table 3) showed no notable performance degradation across various tasks, we agree that ALPT's success generally depends on sample quality.
> Explicitly enumerating and discussing design principles, as demonstrated in our study, should help maintain high-quality samples.

---

### Author Rebuttal · Authors · 2024-08-07

We thank the reviewers for their valuable feedback!
We will update the paper to address the reviewers' suggestions as follows:

* **Additional experiments on math and coding suggested by reviewer aRM4 show that ALPT significantly enhances LLMs' capabilities in various tasks in these domains** (Table F.9 in the newly attached PDF on OpenReview). Combined with the original paper's results, ALPT demonstrates improvement in various LLM capabilities, from reasoning in natural language and NLI to math and coding. We will include these interesting results to strengthen the paper.
* We will include additional experiments on smaller dataset sizes (Table F.10 in the PDF), as suggested by reviewer zCCP, to provide insights on the trade-off between dataset efficiency and performance.
* To make the paper more self-contained, we will include:
    1. Dataset statistics of PLD and baseline corpora (suggested by reviewer zCCP)
    2. The dataset generator workflow (suggested by reviewer BW1m)
* We will emphasize our study's important position in the context of synthetic logic corpora, currently only shown in Appendix A.1, as replied to reviewer BW1m.

---

### Decision · Program_Chairs · 2024-09-25

**Decision:**

Accept (poster)

**Comment:**

This is an interesting work to enhance reasoning ability of LLMs using synthetic rule-based data. By training on the additional data with as continual pretraining, LLMs show better performance in logical reasoning without forgetting the pretrained knowledge. Two of the three reviewers are convinced with the contribution of the paper, while the other reviewer who was against it, didn't participate in the rebuttal.